# Utilizing Weak-to-Strong Consistency for Semi-Supervised Glomeruli Segmentation

| | |
|---|---|
| **Irina Zhang**[1] | YISHUO.ZHANG1@ASTRAZENECA.COM |
| **Jim Denholm**[1] | JIM.DENHOLM@ASTRAZENECA.COM |
| **Azam Hamidinekoo**[1] | AZAM.HAMIDINEKOO@ASTRAZENECA.COM |
| **Oskar Ålund**[2] | OSKAR.ALUND@ASTRAZENECA.COM |
| **Christopher Bagnall**[1] | CHRISTOPHER.BAGNALL1@ASTRAZENECA.COM |
| **Joana Palés Huix**[2,4] | JOANA.PALESHUIX@ASTRAZENECA.COM |
| **Michal Sulikowski**[1] | MICHAL.SULIKOWSKI@ASTRAZENECA.COM |
| **Ortensia Vito**[3] | ORTENSIA.VITO@ASTRAZENECA.COM |
| **Arthur Lewis**[1] | ARTHUR.LEWIS@ASTRAZENECA.COM |
| **Robert Unwin**[3] | ROBERT.UNWIN@ASTRAZENECA.COM |
| **Magnus Söderberg**[2] | MAGNUS.SODERBERG@ASTRAZENECA.COM |
| **Nikolay Burlutskiy**[1] | NIKOLAY.BURLUTSKIY@ASTRAZENECA.COM |
| **Talha Qaiser**[1] | TALHA.QAISER1@ASTRAZENECA.COM |

[1] *Imaging and Data Analytics, Clinical Pharmacology & Safety Sciences, R&D, AstraZeneca, UK*

[2] *Pathology, Clinical Pharmacology & Safety Sciences, R&D, AstraZeneca, Sweden*

[3] *Translational Science and Experimental Medicine, AstraZeneca, UK*

[4] *KTH Royal Institute of Technology, Sweden*

## Abstract

Accurate segmentation of glomerulus instances attains high clinical significance in the automated analysis of renal biopsies to aid in diagnosing and monitoring kidney disease. Analyzing real-world histopathology images often encompasses inter-observer variability and requires a labor-intensive process of data annotation. Therefore, conventional supervised learning approaches generally achieve sub-optimal performance when applied to external datasets. Considering these challenges, we present a semi-supervised learning approach for glomeruli segmentation based on the weak-to-strong consistency framework validated on multiple real-world datasets. Our experimental results on 3 independent datasets indicate superior performance of our approach as compared with existing supervised baseline models such as U-Net and SegFormer.

**Keywords** Chronic kidney disease, semi-supervised learning, computational pathology

## 1. Introduction

Chronic kidney disease (CKD) involves the loss of kidney function over time with little or no reversal. CKD affects more than 800 million people and is expected to become the fifth-highest cause of death worldwide by 2040 (Vanholder et al., 2021). Accurate diagnosis requires pathologists to examine tissue morphology on renal biopsies. This involves identification and quantification of glomeruli, the primary filtration units responsible for filtering plasma. Digitization of biopsies enabled researchers to develop AI models to automatically segment glomeruli on Whole Slide Images (WSIs). However, existing models often generalize poorly on real-world datasets (RWDs) due to the high variability of histopathology images collected from multiple sources.

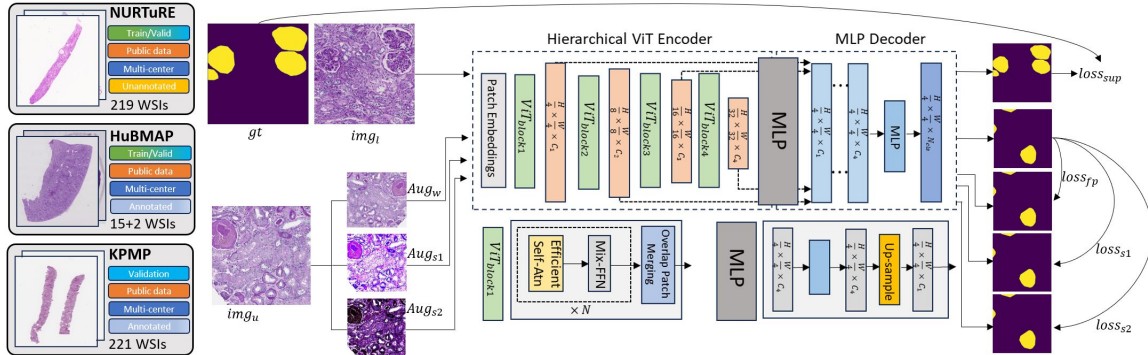

Figure 1: The schematic diagram of our semi-supervised model, contains an encoder and decoder. The training loss was jointly optimized for supervised and unsupervised samples.

*Other work on glomeruli segmentation—* Previous studies investigating glomeruli segmentation using RWDs are relatively scarce. Li et al. (2023) proposed combining contrastive learning and consistency regularization for the detection of glomeruli instead of pixel-level segmentation. Saikia et al. (2023) designed a light-weight U-Net using MLP-based encoders for glomeruli segmentation. However, Schena et al. (2022) suggested AI models analysing renal biopsies trained on limited retrospective data might perform differently on RWDs.

*Our contribution—* We propose a semi-supervised-learning (SSL) approach to improve glomeruli segmentation on renal biopsies from diverse, large-scale RWDs, including: (I) NURTuRE (the National Unified Renal Translational Research Enterprise), the first kidney biobank for CKD collected from 13 nephrology centers across the UK (Taal et al., 2023); (II) KPMP (Kidney Precision Medicine Project), a prospective cohort to investigate the mechanisms of CKD obtained from 9 sites across the US (de Boer et al., 2021); (III)) HuBMAP (Hacking the Kidney Competition) (Howard, 2020) and HuBMAP (Hacking the Human Vasculature Competition) (Howard, 2023), two publicly available renal resection datasets. Our algorithm leverages both labeled and unlabeled data and noticeably outperformed supervised baseline models when validated on external datasets.

## 2. Methods

We adapted UniMatch SSL (Yang et al., 2023) and SegFormer (Xie et al., 2021) to build the framework shown in Figure 1 for glomeruli segmentation on Periodic Acid Schiff (PAS)-stained renal biopsies collected from diverse multi-center real-world cohorts.

**Supervised baseline models**. We compared the performance of the winning solution for HuBMAP – Hacking the Kidney competition (Attention U-Net) (Howard, 2020) and SegFormer (Xie et al., 2021), a light-weight segmentation model based on vision transformers. We chose SegFormer for semi-supervised training in the following steps given its better overall performance.

**Semi-supervised glomeruli segmentation**. In SSL, the core task is to define a valid training strategy to leverage unlabeled data. Sohn et al. (2020) propose a weak-to-strong consistency regularization by using high-confidence predictions made on weakly-augmented image as pseudo labels and train the model to make consistent predictions on strongly-augmented versions of the same image.

| Dataset | Method | Precision | Recall | Dice |
|---------|--------|-----------|--------|------|
| HuBMAP Vasculature | Sup baseline (SegFormer) | 0.95 | 0.545 | 0.685 |
| | Sup baseline (U-Net) | **0.954** | 0.491 | 0.646 |
| | UniMatch$_{SF}$ (the proposed) | 0.874 | **0.761** | **0.814** |
| KPMP | Sup baseline (SegFormer) | **0.947** | 0.622 | 0.751 |
| | Sup baseline (U-Net) | 0.945 | 0.606 | 0.737 |
| | UniMatch$_{SF}$ (the proposed) | 0.93 | **0.647** | **0.763** |
| NURTuRE Labeled | Sup baseline (SegFormer) | 0.822 | 0.528 | 0.643 |
| | Sup baseline (U-Net) | 0.81 | 0.523 | 0.634 |
| | UniMatch$_{SF}$ (the proposed) | **0.823** | **0.573** | **0.675** |

Figure 2: (left) Quantitative results, (right) and a few examples showing qualitative results of the proposed approach as compared to supervised (Sup) baselines.

UniMatch (Yang et al., 2023) expands the perturbation space proposed by FixMatch (Sohn et al., 2020) by training the model to make consistent predictions on unlabeled input images after weak augmentation, feature perturbation, and two different strong augmentations. The expanded multi-stream perturbation approach enables the model to capture comprehensive details between these strong and weak augmented images.

## 3. Results and Discussion

**Datasets: D1-a:** NURTuRE (Taal et al., 2023) contains 13,145 image patches from 198 WSIs, gathered from 11 centers without annotations. **D1-b:** NURTuRE Labeled, 1700 patches from 21 WSIs from the 2 hold-out centers **D2:** KPMP (de Boer et al., 2021), 20,868 patches from 210 annotated WSIs. **D3-a:** HuBMAP Kidney (Howard, 2020), 3706 image patches from 15 annotated WSIs. **D3-b:** HuBMAP-Vasculature (Howard, 2023), 416 image patches extracted from 2 annotated WSIs. All image patches excluding from D3-b were of $1024 \times 1024$ and for D3-b the patch size was $512 \times 512$ at $20\times$ magnification.

**Implementation details:** The models were trained with an initial learning rate of 0.0001 using Stochastic Gradient Descent (SGD) optimizer with a momentum of 0.9. We used a batch size of 10 and trained all the models for 30 epochs. We used D3-a for supervised training and D1-a as unlabeled data in SSL. External validation was performed on D1-b, D2, and D3-b.

**Results:** Figure 2 reports the segmentation results of our approach against the supervised baselines. For supervised baselines, we observed that SegFormer overall performs better than U-Net, even though SegFormer-b1 (13.7M) has only one-tenth of the parameters of U-Net (134.1M), making it more cost-efficient to fine-tune or retrain. For the proposed variant of UniMatch$_{SF}$, we opted to use SegFormer-b1 as encoder and MLP as decoder. Our SSL approach allowed the model to learn from unlabeled images from RWD and perform significantly better on all 3 validation datasets (D1-b, D2, and D3-b) in terms of dice and recall metrics while attaining comparable precision.

This study highlights the significance of SSL approaches for handling multi-center real-world histopathology data. We believe that the presented line of research can be extended to classify different types of glomerular diseases or morphological changes, potentially assisting pathological examination of renal biopsies.

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
