# OpenReview forum: "Utilizing Weak-to-Strong Consistency for Semi-Supervised Glomeruli Segmentation"
_MIDL.io/2024/Short_Papers — MIDL 2024 Short Papers_

### Official Review · Reviewer_27ue · 2024-04-16

**Confidence:** 5
**Final Rating:** 5

**Review:**

The paper presents a semi-supervised learning (SSL) approach to improve glomeruli segmentation in renal biopsies, leveraging diverse and extensive real-world data. The strength of the study lies in its innovative use of SSL techniques to harness both labeled and unlabeled data from significant datasets, including the UK's first kidney biobank, NURTuRE, and the US-based KPMP, along with datasets from the HuBMAP competitions. This approach not only enhances segmentation accuracy but also demonstrates superior performance compared to existing supervised models when tested on external datasets. These aspects position the paper strongly for acceptance, highlighting its potential impact on improving CKD diagnosis and treatment through advanced image analysis techniques.

---

### Decision · Program_Chairs · 2024-04-26

Accept